# Abscisic Acid and Jasmonate Metabolisms Are Jointly Regulated During Senescence in Roots and Leaves of *Populus trichocarpa*

**DOI:** 10.3390/ijms21062042

**Published:** 2020-03-17

**Authors:** Natalia Wojciechowska, Emilia Wilmowicz, Katarzyna Marzec-Schmidt, Agnieszka Ludwików, Agnieszka Bagniewska-Zadworna

**Affiliations:** 1Department of General Botany, Institute of Experimental Biology, Faculty of Biology, Adam Mickiewicz University, Uniwersytetu Poznańskiego 6, 61-614 Poznań, Poland; katarzyna.marzecschmidt@gmail.com; 2Chair of Plant Physiology and Biotechnology, Faculty of Biological and Veterinary Sciences, Nicolaus Copernicus University, Lwowska 1, 87-100 Toruń, Poland; emwil@umk.pl; 3Department of Biotechnology, Institute of Molecular Biology and Biotechnology, Faculty of Biology, Adam Mickiewicz University, Uniwersytetu Poznańskiego 6, 61-614 Poznań, Poland; ludwika@amu.edu.pl

**Keywords:** senescence, phytohormones, abscisic acid, jasmonate, absorptive roots, leaf senescence, microarrays analyses

## Abstract

Plant senescence is a highly regulated process that allows nutrients to be mobilized from dying tissues to other organs. Despite that senescence has been extensively studied in leaves, the senescence of ephemeral organs located underground is still poorly understood, especially in the context of phytohormone engagement. The present study focused on filling this knowledge gap by examining the roles of abscisic acid (ABA) and jasmonate in the regulation of senescence of fine, absorptive roots and leaves of *Populus trichocarpa.* Immunohistochemical (IHC), chromatographic, and molecular methods were utilized to achieve this objective. A transcriptomic analysis identified significant changes in gene expression that were associated with the metabolism and signal transduction of phytohormones, especially ABA and jasmonate. The increased level of these phytohormones during senescence was detected in both organs and was confirmed by IHC. Based on the obtained data, we suggest that phytohormonal regulation of senescence in roots and leaves is organ-specific. We have shown that the regulation of ABA and JA metabolism is tightly regulated during senescence processes in both leaves and roots. The results were discussed with respect to the role of ABA in cold tolerance and the role of JA in resistance to pathogens.

## 1. Introduction

Senescence is a coordinated series of events that begins at a cellular level and then broadens to whole tissues, organs, and in monocarpic plants, the whole organism [1]. In crops, this process usually overlaps with the reproductive phase and might influence reducing crop yield when it is induced prematurely or/and under unfavorable environmental conditions [2]. The onset of senescence may be related to age [3,4] or can be stimulated by internal factors, such as reactive oxygen species (ROS) or changes in phytohormone level, as well as external factors such as photoperiod, temperature, nutrient deficiency, or shading [5,6,7,8]. While the process of senescence appears to be outwardly destructive, the fact that it is a highly regulated process allows plants to relocate a significant portion of valuable nutrients from senescing organs to other tissues so that they can be re-utilized rather than lost [9,10].

One of the most characteristic features of senescence is an increase of catabolic metabolic reactions in relation to anabolic ones [11]. This shift leads to the degradation of individual organelles and eventually the rupture of the tonoplast, protoplast acidification, and cell death [8,12,13,14]. Chloroplasts, which contain approximately 80% of total leaf nitrogen, are one of the first organelles to be degraded [15,16]. Massive degradation of chloroplast proteins as well as other macromolecules initiates the main goal of senescence: remobilization [17]. Contrary to chloroplasts, the nucleus, which is essential to coordinate senescence progression by gene transcription, and mitochondria, which supply energy, remain unharmed up to the last stages of senescence [18,19]. Several studies have indicated that abscisic acid (ABA) [7,20], jasmonic acid (JA) [21,22], ethylene (ET) [23,24,25], and salicylic acid (SA) [26,27] promoted senescence in leaves and flower petals before they were ceased, while cytokinins (CKs) [28,29] delayed this process. This has been confirmed by the analysis of phytohormone levels during senescence, as well as by molecular studies in which the expression of genes related to the biosynthesis of phytohormones was found to increase in senescing organs [30]. The expression of a wide range of genes is also modulated during senescence. Specifically, genes that are up-regulated during senescence are termed *Senescence-Associated Genes* (*SAGs*), while genes that are down-regulated are called *Senescence down-regulated genes* (*SDGs*) [31]. Functional analyses have revealed that among *SAGs* are those genes encoding proteolytic enzymes, ATG proteins, nitrogen-metabolizing enzymes, and several transcriptional factors (TFs) such NAC or MYB [17,32]. In contrast, *SDGs* encode proteins related to photosynthesis or oxidative enzymes, such as catalase [33].

Programmed cell death (PCD) is a cellular process that is often associated with senescence, and such relation was confirmed for leaves and flower petals [8,14,34]. Recent studies have provided evidence that PCD is also involved in the senescence of fine roots [12,13]. Fine, absorptive roots constitute an important component of soil biomass and play a significant role in biogeochemical cycling in forest ecosystems [35,36]. Based on recent studies, it is apparent that fine roots, with a diameter < 2 mm, should not be considered as a homogeneous entity because they include both absorptive and transport roots. The first two to three root orders are classified as absorptive, fine roots. These roots are characterized by high absorption and respiration rates, and they are often colonized by mycorrhizae [13,35,36,37,38]. The life span of fine roots is species-specific and may range from a few weeks to as long as two years [39,40]. However, the average life span of fine roots in *Populus* is typically < 95 days. New insights on root senescence suggest that it is not a passive process but rather is genetically regulated [12,13]. Similar to flower petals and leaves, changes in morphology (color changes, wilting/shrinkage) and cytology (tonoplast rupture, changes in cell shape) occur, including the activation of autophagy. Despite this cursory information, a comprehensive outlook on the senescence of underground organs is still lacking. Considering the total biomass of fine, absorptive roots, a better understanding of senescence and death in these organs is essential to understanding chemical element circulation in forest ecosystems.

In the present study, we report on significant changes that occur in gene expression during the senescence of leaves and fine, absorptive roots in *Populus trichocarpa*. We focused on checking which changes accompanied senescence of two organs that play completely different roles in plant physiology and metabolism, and they also exist under completely different growth conditions. Comparison of these two organs was conducted to determine if the senescence process, despite significant differences in structures among others related to the presence of a photosynthetic apparatus in leaves, followed the same or independent pathways. The results obtained from transcriptomic and physiological analyses as well as microscopic localization provided new information that ABA and JA may contribute to senescence processes in leaves and fine roots and confirmed that the senescence of fine, absorptive roots should be classified as another example of PCD. We suggest that the senescence of both studied organs is a hormonally regulated process, but this regulation in both organs is different, which indicates organ-specific hormonal regulation.

## 2. Results

### 2.1. Changes in Gene Expression During the Senescence of Leaves and Roots

Before carrying out the analyses, morphological and anatomical characteristics of the plant material were performed, as described previously [13]. In addition, basic physiological parameters for leaf senescence, such as chlorophyll level, were examined, and the ratio of fresh mass (FW) / dry mass (DW) was calculated (Appendix A). On this basis, sampling moments related to control variants and senescence stages were estimated. The main senescence-related feature that was taken into account was the change of color. For leaves, it was associated with a decrease in the chlorophyll level (Appendix A); for roots, in addition to color change, shrinkage was also included. The microarray analyses were conducted at three time points for each organ and included viable organs without any symptoms of senescence (LC, RC) as well as early (RS1, LS1) and advanced stages of senescence (RS2, LS2) (Figure 1).

Microarray analyses revealed significant changes in gene expression during the course of senescence of roots and leaves.

In roots, a total of 1898 differentially expressed genes (DEGs) were identified (One-way ANOVA corrected *p*-value cut-off = 0.001, post-hoc Tukey HSD and Benjamini–Hochberg correction, fold change ≥ 2) in the three stages of senescence that were analyzed. Most of the DEGs exhibited down-regulation in the early and late stages of senescence. A total of 924 DEGs were down-regulated in RS1 and 1169 in RS2. In contrast 556 DEGs were up-regulated in RS1 and 692 in RS2 (Figure 2A). 

Most DEGs were either down- or up-regulated in both RS1 and RS2 stages (Figure 2A); however, there was a significant number of DEGs that either decreased or increased in their expression only in the final stage (RS2) of senescence. A total of 1500 DEGs were annotated, and 101 clusters and 77 functional categories were identified using the DAVID database [41,42]. The majority of the annotated genes encoded proteins located in extracellular regions (35 genes), the cytosol (34 genes), cell wall (31 genes), or plant-type cell wall (22 genes), or as an integral component of the plasma membrane (27 genes) (Appendix A). Gene Ontology (GO) enrichment analysis revealed that the most abundant categories were carbohydrate metabolic process (33 genes), ROS, and oxidative stress (response to oxidative stress, 15 genes; hydrogen peroxide catabolic process, 14 genes), cell wall (cell wall organization, 15 genes; plant-type cell wall organization, 14 genes; pectin catabolic process, 14 genes; xyloglucan metabolic process, 8 genes; cell wall biogenesis, 8 genes; cellulose catabolic process, 5 genes; xylan biosynthetic process, 4 genes; S-adenosylmethionine biosynthetic process, 3 genes; S-adenosylmethionine cycle, 2 genes), and microtubules (microtubule-based movement, 18 genes; microtubule-based process, 11 genes) (Appendix A). Notably, many genes (90) associated with phytohormones were also identified (Figure 3A).

In leaves, a total of 1348 DEGs (One-way ANOVA corrected *p*-value cut-off = 0.005, post-hoc Tukey HSD and Benjamini–Hochberg correction, fold change ≥ 2) were identified over the course of the senescence process. The majority of the DEGs were up-regulated, 798 in LS1 and 1025 in LS2 stage, while 170 DEGs were down-regulated in LS1 and 303 in LS2 (Figure 2B). Of all DEG, only a small group of genes have been identified as common to both organs (Figure 2C). A total of 1063 DEGs were annotated, and 63 clusters and 42 functional categories were identified using the DAVID database [41,42]. Most of the annotated DEGs encoded proteins assigned, among others, to the nucleus (114 genes), cytoplasm (81 genes), and intracellular (27 genes) categories (Appendix A). GO enrichment analysis revealed that the most abundant enriched categories were those related to DNA-templated transcription (40 genes), protein degradation (proteasome-mediated ubiquitin-dependent protein catabolic process, 15 genes; protein ubiquitination involved in ubiquitin-dependent protein catabolic process, 13 genes; ubiquitin-dependent protein catabolic process, 8 genes), signaling (intracellular signal transduction, 11 genes; small GTPase-mediated signal transduction, 9 genes; Wnt signaling pathway, 4 genes), vesicle-mediated transport (8 genes), endocytosis (4 genes), as well as a plethora of categories related to phytohormones (77 genes) (Figure 3B, Appendix A).

### 2.2. Genes Associated with Phytohormones

Genes associated with phytohormone pathways were identified among the DEGs of both organs using Gene Ontology (GO) within the Biological Process (BP) category. These included genes related to abscisic acid (ABA), jasmonic acid (JA), brassinosteroids (BRs), cytokinins (CKs), auxin (IAA), ethylene (ET), gibberellin (GA), and salicylic acid (SA) (Figure 4A–D, Appendix A). Further analyses focused on genes related to ABA and JA in both organs since the microarray analysis indicated significant changes in the expression of ABA- and JA-related genes and because these two phytohormones are known as positive regulators of leaf senescence. 

#### 2.2.1. ABA-Related Genes

In roots, genes associated with ABA belonged to three subcategories within the GO BP, with “response to ABA” containing the greatest number of DEGs (Figure 4A). A total of 28 genes fell into this subcategory of BP, among which 12 were up-regulated in RS1 and 10 in RS2 (Figure 3A, Figure 4A; Appendix A). Concomitantly, 16 genes in this subcategory were down-regulated in RS1 and 18 in the RS2 stage of senescence. Among the identified up-regulated DEGs were genes encoding proteins related to stress response, including cold stress—*WCOR413* (POPTR_0004s15610) and *COR413IM1* (POPTR_0001s34410)—or oxidative stress—the precursors of ferritin (POPTR_0008s07270, POPTR_0010s19190). The down-regulated DEGs included several genes encoding aquaporin-related proteins (POPTR_0008s04430, POPTR_0010s22950, POPTR_0004s18240, POPTR_0009s01940). Several DEGs encoding proteins involved in signaling such as kinases (POPTR_0017s02820, POPTR_0001s23030, POPTR_0010s00490) or phosphatase (POPTR_0010s20720) were also found to be down-regulated (Appendix A).

In leaves, genes associated with ABA were placed in six subcategories of GO BP. Similar to fine roots, the GO BP subcategory “response to ABA” contained the largest number of DEGs; however, “ABA-activated signaling pathway” was also significantly represented (Figure 4B). Among the 33 ABA-related DEGs, 25 were up-regulated and 8 were down-regulated during both stages (LS1 and LS2) of senescence (Figure 3B; Figure 4C). Among the genes up-regulated by senescence were genes encoding transcription factors (TFs), e.g., MYB (PtrMYB168 POPTR_0019s11090, POPTR_0013s1136), bZIP (POPTR_0014s02810), and NAC (NAC034 POPTR_0005s20240, NAC052 POPTR_0003s16490). Up-regulated expression was also observed for genes related to carbohydrate metabolism (POPTR_0001s23090, POPTR_0001s23060), lipid metabolism (POPTR_0001s14290), as well as genes associated with the protein degradation process (POPTR_0004s17940, POPTR_0012s09300, POPTR_0005s27480). Notably, genes encoding proteins associated with ABA signal transduction, such as phosphatase 2C (POPTR_002s00880, POPTR_001s25200, POPTR_0010s20720) or kinases—SNF1-related protein kinase KIN10 (POPTR_0004s11500) and calcium-dependent protein kinase 1 (POPTR_0019s00630)—were also observed. Similar to the situation in fine, absorptive roots, down-regulation of DEGs encoding an aquaporin-related protein (POPTR_0009s13890) and a precursor of ferritin (POPTR_0010s19190) were also observed (Appendix A).

#### 2.2.2. JA-Related Genes

In roots, genes associated with JA were placed in three subcategories of GO BP, with “response to JA” and “JA-mediated signaling pathway” being the subcategories containing the greatest number of DEGs (Figure 4B). A total of 14 DEGs, among which 6 were up-regulated and 8 were down-regulated, were identified in both stages of root senescence (Figure 3B; Figure 4B). Increased expression of DEGs was identified for genes encoding MYB TF (POPTR_0001s19070), SWI/SNF complex-related (POPTR_0017s01140), ERECTA-like protein (POPTR_0004s16110), and mitochondrial import inner membrane translocase subunit Tim17/Tim22/Tim23 family protein (POPTR_0001s24670). Down-regulated DEGs included, among others, those related to flavonoid metabolism (POPTR_0014s14200, POPTR_0003s11900, POPTR_0001s14310, POPTR_0001s08410) and lipid metabolism (POPTR_0001s15530) (Appendix A).

In leaves, DEGs associated with JA were placed in the same three categories as the DEGs for fine roots. However, in the case of leaves, the subcategory of GO BP “JA biosynthetic pathway” contained the second greatest number of DEGs (Figure 4D). A total of 12 genes were identified, 8 and 7 of which were up-regulated in the LS1 and LS2 stages, respectively (Figure 3B; Figure 4D). Concomitantly, 4 DEGs were down-regulated in LS1 and 5 in LS2. The greatest increase in expression was observed for DEGs encoding transcriptional factors, such as an ethylene-responsive transcription factor RAP2-3 (POPTR_0010s00900), TBF1 (POPTR_0001s24840), and MYBs (PtrMYB180 POPTR_0013s11360 and PtrMYB168 POPTR_0019s11090). In addition, increased expression was observed for a gene similar to JAZ1, which encodes a jasmonate ZIM-domain protein (POPTR_0006s14160) (Appendix A).

These results indicate that the process of senescence in both leaves and fine, absorptive roots is not a passive process but rather a genetically regulated process that is accompanied by significant changes in gene expression, including a large group of genes associated with phytohormone synthesis and signaling pathways. 

### 2.3. ABA, JA, and MeJA Levels During Senescence

Based on microarray analyses and literature data about the role of ABA and JA during leaf senescence, further analyses were carried out for these two phytohormones in order to check whether organs with different functions used similar mechanisms or affected the metabolism of these compounds to the progression of senescence process.

Quantitative analyses of ABA, jasmonic acid (JA), and methyl jasmonate (MeJA) revealed significant differences in the concentration of these phytohormones during the senescence of leaves and fine absorptive roots (Figure 5, Figure 6). The results of changing the phytohormone concentration were statistically significant (*p* < 0.05) (Appendix A). The level of ABA in roots increased during senescence, with the highest level observed in the second stage of senescence (RS2) (Figure 5A). In contrast, the highest level of ABA in leaves was observed in LS1, when leaves were yellowing (Figure 5B).

Quantitative analyses of jasmonates (JA and MeJA) were also conducted. The concentration of both JA and MeJA in both organs increased during senescence, with the highest levels observed in the second stage of senescence (LS2, RS2) (Figure 6A,B). All of the observed changes in jasmonate levels were statistically significant at a *p*-value < 0.05. 

### 2.4. Immunolocalization of ABA and JA During Senescence

Based on the significantly increased concentrations of ABA and jasmonate in both leaves and fine, absorptive roots in the different stages of senescence, immunohistochemical detection of ABA and JA was conducted to determine their cellular and tissue distribution.

#### 2.4.1. ABA Localization

In roots, ABA was localized in the rhizodermis, cortical parenchyma cells, and vascular tissue (mainly in xylem) of control, viable roots (RC) (Figure 7A,B). A signal for ABA was detected in the peripheral cytoplasm of cortical parenchyma cells, which had a large central vacuole (Figure 7A,B; arrows). In the first stage of root senescence (RS1), the ABA signal was observed in the same tissues as in RC, but the cellular distribution was different in cortical parenchyma cells. In this case, the signal was not only concentrated in of the peripheral cytoplasm, but it was also evident in the vacuole (Figure 7C,D; arrowheads). The majority of cortical parenchyma cells were folded and irregular in shape in RS2, which made determining the precise distribution of ABA in these cells problematic. Nevertheless, an intense fluorescent signal was detected in the folded cortical parenchyma cells, as well as in xylem tracheary elements (Figure 7E,F).

In leaves, the highest intensity of ABA signal in green, control leaves (LC) was detected in vascular bundles (Figure 8A–C; arrowheads). A weak signal was also observed in epidermal cells (Figure 8A,B), as well as in a few mesophyll cells (mainly in the palisade layer) (Figure 8C). Chloroplasts were readily defined due to the strong autofluorescence emitted by chlorophyll (Figure 8A–C). ABA in yellowing leaves (LS1) was detected in the majority of palisade and spongy mesophyll cells (Figure 8D–F). The signal was localized in the cytoplasm of these cells and sometimes in spherical spots inside the vacuole (Figure 8F, arrow). Chlorophyll autofluorescence was still visible (Figure 8D–F), whereas it decreased significantly in the second stage of senescence (LS2) (Figure 8G–I). The distribution of the ABA signal in LS2 leaf tissues, however, was different than it was in LS1; the signal was detected as different-shaped spots in the majority of mesophyll cells (Figure 8G–I, arrowheads). 

#### 2.4.2. JA Localization

In roots, the JA signal in viable, control roots (RC) was concentrated mainly in phloem cells (Figure 9A,B). The JA signal was also detected in several cortical cells (Figure 9A,B). In the first stage of root senescence (RS1), the pattern of the JA signal was quite different; the JA was detected in the majority of cortical parenchyma cells, but in contrast to the signal in RC samples, the JA signal was observed throughout the cell rather than just in the peripheral cytoplasm of cells with large vacuoles (Figure 9C,D; arrowheads). In the second stage of senescence (RS2), the anatomical structure of roots was greatly degraded by the senescence process; however, the intensity of the signal was still very high and localized mainly in the folded cortical parenchyma cells (Figure 9E,F) as well as in phloem cells (Figure 9E).

In leaves, the JA signal in green, control leaves was weak and was detected in cells within the vascular bundle (Figure 10A–C), the epidermis (Figure 10A), and a few mesophyll cells (Figure 10C). In mesophyll cells, JA was localized mainly in close proximity to chloroplasts (Figure 10C, arrow). In yellowing leaves (LS1), the JA signal was observed in the majority of cells (Figure 10C–E). The distribution of the signal in mesophyll cells, however, was different than in the LC stage. In addition to areas near the chloroplast, the signal appeared to be distributed throughout the cytoplasm (Figure 10D,F; arrowheads). Notably, an intense signal was still visible in the cells within the vascular bundles (Figure 10E). In the second stage of leaf senescence (LS2), JA was detected in the same tissues as in LS1, but the pattern of localization was different. In this case, the JA signal was observed as spherical spots within palisade mesophyll cells (Figure 10G,H, arrows) or it was concentrated in the peripheral cytoplasm against the cell wall of some cells (Figure 10I). As in the previous stage, the JA signal was still observed within cells of the vascular bundles (Figure 10G). Chlorophyll autofluorescence was very low in the LS2 stage of leaf senescence (Figure 10G–I). 

Negative control reactions had an undetectably low signal relative to the standard reactions (Appendix A). The results obtained for ABA and JA indicated that these phytohormones might play an important role in the senescence process in both leaves and fine, absorptive roots. This was indicated by changes in the expression of many genes associated with ABA and JA, as well as quantitative analyses of phytohormone concentrations. In addition, the localization of the studied phytohormones showed that signal distribution accumulated in tissues with visible signs of senescence (shape changes, lowering of the chlorophyll autofluorescence level).

## 3. Discussion

Over the past decade, extensive genetic, molecular, and physiological studies have revealed the intricate network controlling the process of senescence. The majority of studies have focused on elucidating the senescence process in leaves, which are a fundamental site for capturing energy through photosynthesis [1,19,43]. The onset of leaf senescence is definitely easier to monitor than in the case of the belowground organ. Moreover, in chloroplasts there is a large pool of nitrogen stored; therefore, these organs represent an ideal model to study nutrient remobilization, which is one of the critical aspects of senescence [9,17,44]. Much less attention has been focused on other organs. Recent studies have drawn attention, however, on the senescence of other ephemeral organs, such as fine, absorptive roots [12]. The senescence process in fine roots has many similarities to leaf senescence, among which ultrastructural changes and/or the activation of autophagy-related mechanisms that have been emphasized [13]. Fine, absorptive roots also possess a high concentration of nitrogen, which, along with their large total biomass, provide an interesting model for studying the remobilization of nutrients [45]. There is still, however, a lack of information pertaining to the senescence process of underground organs, especially at the physiological and molecular levels.

In the present study, details regarding the genetic and phytohormonal regulation of the senescence of roots have been presented, along with a comparison of the senescence process in leaves vs. fine absorptive roots. Microarray data indicated that the senescence process of both of the studied organs is accompanied by significant changes in gene expression. GO classification of the DEGs revealed a plethora of genes associated with phytohormones. Previous studies have documented the importance of plant hormones in the regulation of leaf senescence [8,26,30]. Phytohormones can either act as inhibitors or accelerators of senescence. Many of the identified DEGs were those related to ABA. This phytohormone, in addition to its significant role in developmental processes or responses to environmental stresses [46,47,48,49], is also a well-known positive regulator of plant organ senescence [7,20]. The relationship between ABA and leaf age has been known for a long time. In fact, an increase in the level of ABA in yellowing leaves was first noted in the 1980s [50]. In addition to the effect of increasing levels of endogenous ABA [51,52,53,54], exogenous application of ABA has also been shown to accelerate the yellowing of leaves [53,55]. In the present study, a higher level of endogenous ABA in yellowing leaves of *Populus trichocarpa* was also observed. However, the same relationship was observed in senescing roots. Therefore, based on previous studies of leaves and flower petals [56,57,58], and the results observed in the present study, we suggest that the ABA similarly as in other ephemeral organs might also contribute to senescence process in fine roots.

Since ABA also plays a role in several physiological processes in non-senescing organs, it was not surprising that ABA was also detected in viable fine roots and leaves. The immunohistochemical detection of ABA in non-senescent organs, however, was mainly localized to the vascular tissue. The vascular localization may have been associated with ABA transport, which can be transported in both xylem (from roots to shoots) and phloem (from leaves to roots) [59]. An accumulation of ABA was observed in both the peripheral cytoplasm and central vacuole of cortical parenchyma cells of senescing roots. Earlier, such localization in mesophyll cells has also been described in *Arabidopsis* where an abscisic acid glucosyl ester (ABA-GE) was stored in the vacuole, and in response to abiotic stress, ABA-GE can be rapidly converted to the free form of ABA using vacuolar b-glucosidases [60]. In our study, vacuolar localization was not clearly visible in leaves where the ABA signal was primarily detected in the cytoplasm of cells. In the latter stage of senescence (RS2), the structure of cortical parenchyma cells in roots was highly disrupted, making the determination of ABA localization difficult. Nonetheless, a strong signal was observed in the vascular cylinder and within folded cortical parenchyma cells. In contrast, a much lower signal was observed in leaves in the latter stage of senescence (LS2), where the signal appeared as small spherical spots in the cytoplasm of both palisade and spongy mesophyll cells. In LS2, chlorophyll autofluorescence was barely visible, suggesting that the majority of these organelles had been degraded. These observations are in agreement with recent findings on the role of ABA in chlorophagy [61,62]. In that study, the stromule number increased in response to an ABA treatment, whereas treatment with a specific inhibitor of ABA synthesis prevented the formation of stromules by mannitol [61]. Moreover, it has been suggested that proteins belonging to kinase family (SnRK2, CK2), which are activated by the ABA signaling pathway, might participate in phosphorylation of chloroplast membrane proteins or the ATG proteins and activate chlorophagy [62,63].

The analysis of gene expression during senescence, especially ABA-related gene expression, did not reveal many similarities between leaves and fine roots. In fact, the majority of the DEGs were organ-specific. A previous comparative transcriptomic analysis of senescing flower petals and leaves also noted a lack of similarity in the transcriptome of these two organs [64]. In the present study, up-regulated DEGs encoding transcriptional factors (TFs) related to ABA, such as MYB, bZIP, and NAC, were identified in senescing leaves. Similar results were reported in a transcriptomic analysis of senescing leaves of other species, further confirming that senescence is highly regulated by multifold networks [11,65,66,67,68]. The up-regulation of TFs, especially NAC family TFs, during organ senescence has also been documented in numerous crop species [16]. The NAC factor AtNAP has been demonstrated to play a crucial role in the integration of abscisic acid (ABA) signaling and leaf senescence. AtNAP binds to the promoter of phosphatase 2C (PP2C) family genes and activates their expression. One of those PP2C genes encodes a SAG113 protein, which inhibits stomatal closure and, thus, promotes water loss and accelerates leaf senescence [69]. We also detected three genes in senescing leaves of *Populus* encoding phosphatase 2C proteins that are associated with ABA, among which two were up-regulated. Collectively, the data suggested that the PP2C identified in our transcriptome analysis could be involved in the ABA-dependent regulation of leaf senescence. 

Among the up-regulated DEGs related to ABA that were identified in senescing roots, two genes were associated with cold acclimation: *COR413* and *COR314*. The up-regulation of these genes can be induced by both environmental conditions (low temperature) and ABA [70]. COR proteins have been implicated in increasing plant tolerance to low temperature by affecting the metabolism of fatty acids, sugars, and purines. In age-related developmental senescence, preparation for cold may be one of the factors that induces the senescence of fine roots and the up-regulation of genes to enable the acquisition of increased low-temperature tolerance. Cold tolerance would enable the organs (fine roots) to complete the process of senescence and remobilize their nutrients to storage organs rather than just die outright due to low temperatures. Down-regulated genes related to ABA were also identified in both leaves and fine roots, including a gene encoding a PIP aquaporin, which plays a key role in radial water transport in roots and leaves and maintains water homeostasis during the plant response to environmental stress [71]. The expression of aquaporin genes is regulated by a variety of factors, including the concentration of ABA. Jang et al. [72] demonstrated that the expression of the majority of studied *PIP* genes was up-regulated in response to ABA treatment, while several *PIP* genes were down-regulated in response to cold treatment. Throughout the course of our study, plants were exposed to decreasing temperatures that typically occur in autumn and may generally cause a decrease of pressure and reduced sap flow [73]. It is plausible that these conditions may have induced the down-regulation of genes encoding aquaporins, despite the presence of a high level of ABA. Our results indicate that ABA content is tightly regulated during senescence in leaves and roots, possibly at the transcriptional level, and its accumulation may contribute to senescence processes in both organs. We suggest that ABA may increase cold tolerance in fine roots, while it acts as a signal molecule in leaves. As a result, it likely induces the expression of a variety of TFs that contribute to the coordination of several physiological processes, such as the regulation of water loss via the regulation of stomatal aperture.

In addition to ABA-related genes, numerous JA-related genes were also identified in our microarray analyses. JA not only plays a role in the adaptation of plants to biotic and abiotic stresses and the regulation of several developmental events (root inhibition, anthocyanin accumulation, trichome initiation, male fertility, etc.). It also plays a role in the positive regulation of senescence [21,22,74,75,76]. The first report documenting that jasmonate affects senescence was the observation that treatment with methyl jasmonate (MeJA) resulted in a rapid loss of chlorophyll [77]. Although this observation was made a considerable time ago, the molecular mechanisms underlying leaf ageing are still not fully understood. Similar to *Arabidopsis* [22], a significant increase in JA and MeJA levels was observed in senescing leaves of *P. trichocarpa*. The impact of JA on senescence of other ephemeral organs, however, is not so clear. Exogenous application of JA to flower petals of *Phalaenopsis* promoted senescence [78]. In contrast, endogenous levels of JA did not increase during the senescence of *Lilium* flower petals [79], and JA levels decreased while MeJA levels increased in senescing cotyledons of *Ipomoea nil* [80]. There are still insufficient data to confirm the promotive role of JA in root senescence; however, our results indicating variations of JA and MeJA content in senescent roots of *Populus* suggest that these phytohormones can be a metabolic signature of senescence. The localization of JA in viable organs was concentrated primarily in the vascular bundle and might be due to several factors other than senescence, such as the synthesis of JA [81], modulations of the level of transfer cell wall ingrowths in the phloem [82], or JA transport [83]. The highest JA signal in senescing roots was observed in the cortical parenchyma cells, in which significant changes associated with senescence, such as the presence of autophagic-related structures and changes in cell shape, were already observed in RS1 [13]. Ultrastructural analyses also revealed the presence of microorganisms inside cells during the latter stages of senescence (RS2) [12,13]. Thus, the high concentration of JA in senescent roots may have been related to protecting these organs from pathogens, allowing the progress of senescence and, more importantly, nutrient remobilization instead of rapid death. Jasmonates have been reported to induce defense responses against microorganisms that cause plant diseases [84,85,86]. In this regard, *fad3–2, fad7–2*, and *fad8* mutants in *Arabidopsis*, in which JA accumulation is disrupted, exhibited roots that were more susceptible to root rot caused by a fungal root pathogen than the roots of wild-type plants. Notably, exogenous application of MeJA reduced this effect [85].

As found with ABA-related DEGs, the microarray analysis conducted in the present study did not identify JA-related DEGs that were common to both leaves and fine roots. In fact, there was a distinct lack of a significant number of JA-related DEGs in fine roots related to the course of senescence. Increased expression of a gene encoding an ERECTA protein, which functions in the regulation of immune responses and resistance to pathogens, however, was documented [87,88]. Collectively, these results, along with a previous report [12], appear to indicate that JA is not directly involved in the senescence process in roots. JA, however, may indirectly affect senescence by modulating the resistance response to pathogens that would prevent the rapid death of roots due to pathogen invasion. Consequently, this would allow sufficient time for root cells to complete the senescence processes in a prescribed manner, including remobilization, autophagy, etc. The expression of genes encoding MYB TFs was noted in both leaves and roots. MYB TFs are a large family of proteins that are a crucial element in regulatory networks controlling plant development, metabolism, and responses to biotic and abiotic stresses [89]. The up-regulation of several MYB proteins was documented in senescing leaves of *Arabidopsis* [90] and *Solanum* [91]. MYB TFs are also involved in the jasmonate signaling cascade by interacting with JAZ proteins [76,92], which are known as repressors of JA-responsive genes. The function of JAZ proteins in leaf senescence and the regulation of cell death during host and non-host interactions, however, has also been postulated [93,94]. Our results indicate that jasmonate might play an important role in the direct or indirect regulation of senescence in both leaves and roots; however, similar to ABA, the regulatory effect differs in the two organs. We suggest that jasmonate might be involved in the response to biotic stress in senescing roots, while its role in leaves seems to be more complex and overlap with the regulation of several other processes through the influence of jasmonate on activation of TFs. However, confirmation of the exact role of JA in the senescence of plant organs requires further research.

Summarizing, the senescence of plant organs involves an intricate network of episodes that, despite considerable research, still remains insufficiently understood. Particularly enigmatic is the senescence of underground ephemeral plant organs. Thus, in the current study we primarily focused on examining the senescence process in fine absorptive roots. We also sought to determine if the underlying mechanism regulating senescence was universal for all plant organs by comparing the senescence of fine roots with leaf senescence. Our results indicated that the senescence of both organs was accompanied by significant changes in gene expression. Many of the identified DEGs in both organs were associated with a variety of phytohormones. The quantitative analyses of the senescing organs also revealed that the levels of ABA, JA, and MeJA increased in both leaves and roots during senescence. Despite these similarities, however, our analysis of phytohormone-related genes indicated that the function of ABA and jasmonates may differ in the two organs during senescence. We suggest that phytohormones in roots do not directly regulate the progression of senescence but rather act indirectly by regulating other processes such as cold acclimatization (ABA) and resistance to soil microorganisms (JA). Although these regulatory processes are not directly associated with the progression of senescence, they are essential to prevent premature cell death and allow senescence and the processes associated with it (such as remobilization) to occur in a prescribed manner rather than be terminated by premature cell death. In contrast, ABA and JA appear to have a direct role in the regulation of leaf senescence. Our study suggests that the regulatory effect of phytohormones on senescence is organ-specific. The exact mechanism regulating the senescence of leaves, and especially fine, absorptive roots, is still not well understood and should be a topic of future research.

## 4. Materials and Methods

### 4.1. Plant Material and Growth Conditions 

All experiments were performed on *Populus trichocarpa* (Torr. & A. Gray ex Hook.). Seeds were germinated on 1% agar. Seedlings (about 1–2 cm in length) were planted in soil in a seed-starting system. Plants were grown in a growth chamber (Conviron GR96) at 18 °C day/14 ℃ night temperature and a 16 h day/8 h night photoperiod. After two months, plants were transferred to rhizotrons as described in Wojciechowska et al. [95]. The material for the study was sampled during the first vegetative season at three time points based on morphological markers of senescence. For leaves, senescence-related stages were distinguished based on chlorophyll content. Chlorophyll measurements have been performed using a CCM-200 plus Chlorophyll Content Meter (Opti-Sciences, Hudson, NH, USA) in two places for one leaf (on both sides of each leaf blade analyzed). Such measurements were performed for 30 plants each time, throughout the vegetative season. The average result for green leaves (LC) was defined as 100% chlorophyll content. In yellowing leaves (LS1), material was collected in which the chlorophyll content dropped about 40%, and for yellow leaves (LS2) about 60% (Appendix A). For roots, senescence-related stages were also distinguished based on changing their color. Moreover, a viability test, as well as anatomical and cytological analyses, confirmed that color change was associated with the senescence process. The senescence-related stages have been broadly described in Wojciechowska et al. [13] and are presented in Figure 1.

### 4.2. Microarray Analysis

Total RNA was extracted from three biological replicates of leaves and roots using an RNeasy Plant Mini kit (Qiagen, Germantown, MD, USA). RNA quantity and quality were assessed using a NanoDrop1000 (Thermo Fisher Scientific Inc., Waltham, MA, USA). cDNA synthesis and microarray hybridization to an Affymetrix GeneChip Poplar Genome Array (A-AFFY-131) were performed according to the provided Affymetrix protocol. A complete microarray dataset was submitted to the Gene Expression Omnibus database (accession number GSE143559). The raw image data from a total of three A-AFFY-131 arrays were normalized with Robust Multi-Array Average (RMA). The normalized data were statistically analyzed using GeneSpringGX7 13.1 (Agilent Technologies Inc., Santa Clara, CA, USA) software. Data were subjected to a one-way ANOVA with a corrected *p*-value cut-off = 0.05 and a Benjamini–Hochberg correction. DEGs were annotated using Phytozome JGI database, BLAST, Ensembl Genome, and KAGIANA. Heatmaps were generated using MATLAB (The MathWorks Inc., Natick, MA, USA). The Venn diagram of DEGs was drawn by VENNY2.1 (https://bioinfogp.cnb.csic.es/tools/venny/).

### 4.3. Measurement of Endogenous Abscisic acid (ABA) and Jasmonates (JA and MeJA)

GC-MS was used to determine the concentrations of endogenous ABA, JA, and MeJA. Material for phytohormone concentration measurement was collected from nine plants, and measurements were performed in three technical replicates. After the plant material (~0.5 g) was homogenized, ABA and jasmonate levels were measured using the optimized protocols described by Wilmowicz et al. [96,97]. GC-MS-SIM was performed by monitoring *m*/*z* 134, 162, and 190 for measuring endogenous ABA and 138, 166, and 194 for [6-^2^H_3_] ABA according to the method described by Vine et al. [98]. GC/MS-selected ion monitoring was used to measure jasmonates by monitoring *m*/*z* 193, 195, 224, and 226. Statistical analyses (ANOVA with a corrected *p*-value = 0.05 and Tukey’s post-hoc test) were performed using Statistica 12.0 software (StatSoft Poland Inc., Tulsa, OH, USA).

### 4.4. Immunolocalization of JA and ABA

Samples of roots and leaves were fixed in 3% (*v*/*v*) *N*-(3-dimethylaminopropyl)-*N*′-ethylcarbodiimide hydrochloride (EDAC) for 2h and a mixture of 2% glutaraldehyde (pH 6.8; Polysciences, Warrington, USA) and 2% (*v*/*v*) formaldehyde (pH 6.8; Polysciences, Warrington, USA) for 12 h at 4 °C. The fixative was then discarded, and the samples were rinsed three times in 1x PBS (phosphate-buffered saline) (Sigma, St Louis, MO, USA) buffer. Leaf samples (5 × 5mm) were sectioned (30 µm) using a vibratome Leica VT 1200S (Leica Biosystems, Nussloch, Germany), while root samples were dehydrated in a graded ethanol series (10%–100%) and then infiltrated and embedded in Paraplast Extra (melting point, 57.8 °C; Sigma, St Louis, MO, USA). Root samples were sectioned (20 μm) using a Leica RM2265 (Leica Biosystems, Nussloch, Germany) rotary microtome. JA was localized using primary anti-JA rabbit antibodies (Agrisera, Sweden, catalogue number AS11 1799) at a 1:500 dilution. ABA was detected using a primary, anti-ABA rabbit antibody (Agrisera, Sweden, catalogue number AS09 446) at a 1:500 dilution. Immunolocalization assays were performed as described by Wojciechowska et al. [13]. Results of the localization were viewed and recorded with a Leica TCS SP5 confocal microscope (Leica Biosystems, Nussloch, Germany) using the following lasers: 405 diode-emitting light at a wavelength of 405 to observe chlorophyll fluorescence (in leaves) or cell wall fluorescence (in roots to observe the shape of the cells) and an argon laser-emitting light at a wavelength of 488 to observe fluorescence of Alexa 488 (ABA, JA). Negative control reactions consisting of samples processed without exposure to the primary antibodies were utilized for both ABA and JA (Appendix A).

## Figures and Tables

**Figure 1 ijms-21-02042-f001:**
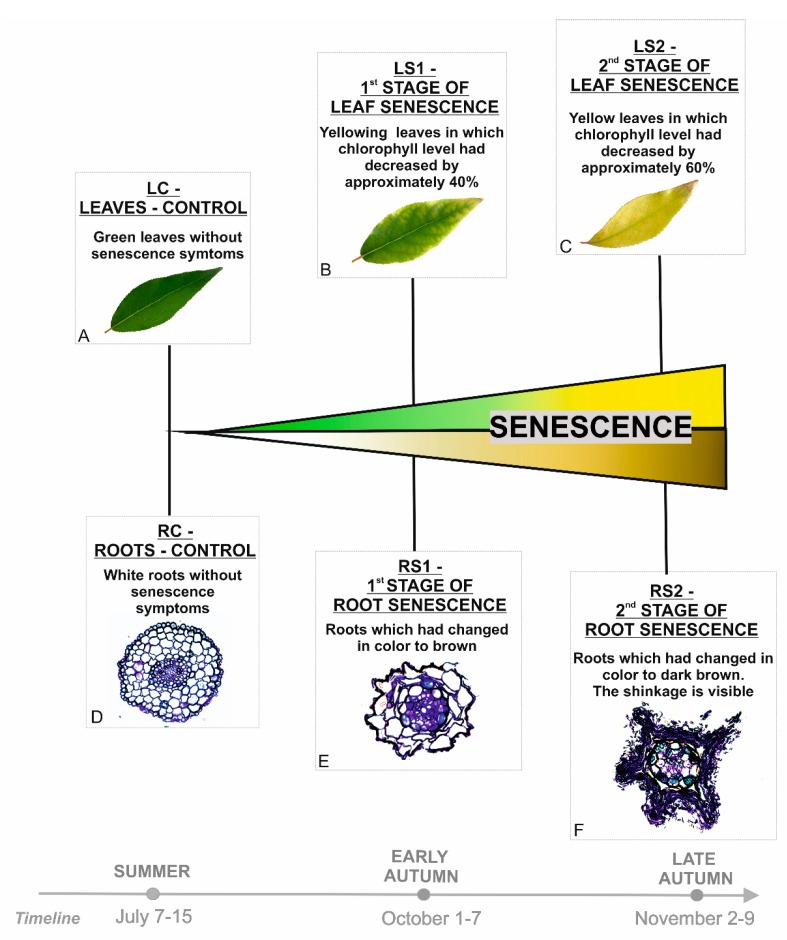
Stages of leaves (**A**–**C**) and fine, absorptive roots (**D**–**F**) selected for analyses based on morphological and anatomical changes observed during the vegetative season.

**Figure 2 ijms-21-02042-f002:**
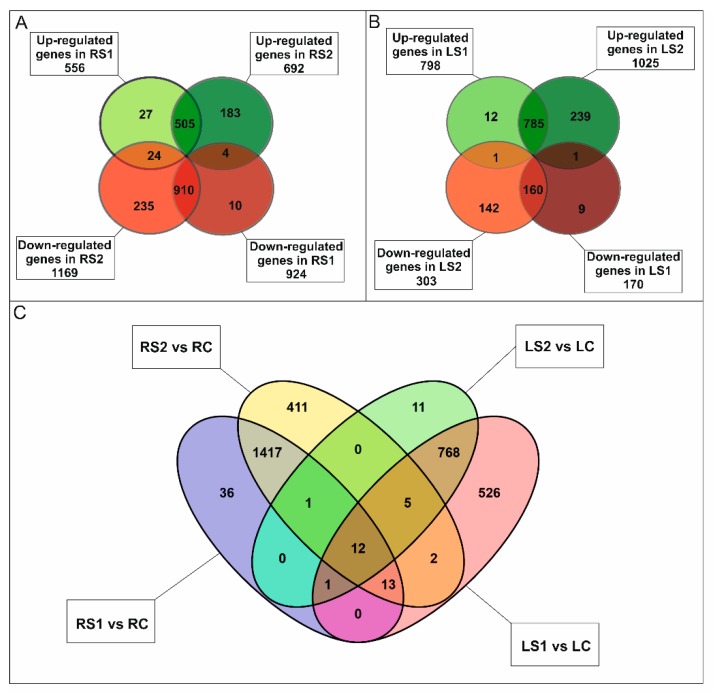
Transcriptomic analysis of fine, absorptive roots and leaves during senescence. (**A**) Venn diagrams showing the expression pattern of 1898 statistically significant differentially expressed genes (DEGs) in fine roots at two time points during senescence and the overlap in expression among them. (**B**) Venn diagrams showing the expression pattern of 1348 statistically significant genes in leaves at two time points during senescence and the overlap in expression among them. (**C**) Venn diagram showing the expression pattern of DEGs in roots and leaves at two time points during senescence and the common genes for both organs.

**Figure 3 ijms-21-02042-f003:**
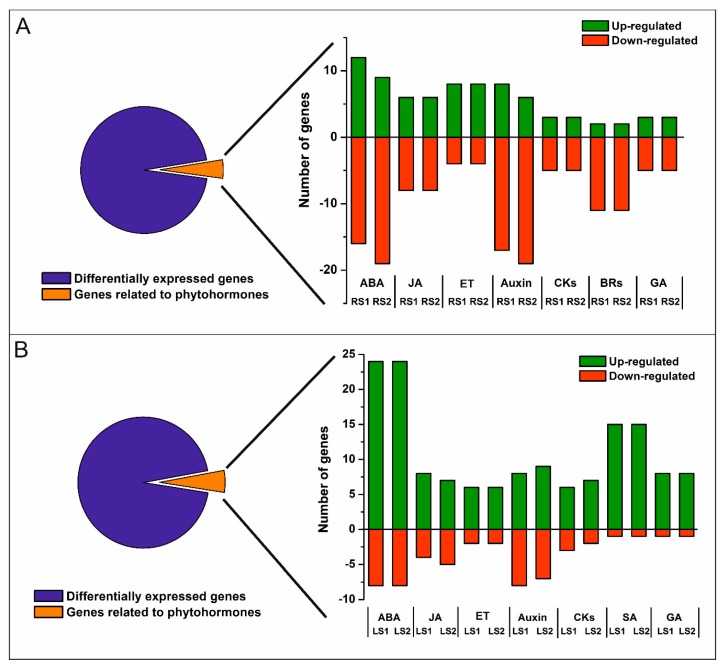
Number of up- and down-regulated phytohormone-related genes in roots (**A**) and leaves (**B**). Abbreviations: ABA, abscisic acid; JA, jasmonic acid; ET, ethylene; CKs, cytokinins; BRs, brassinosteroids; SA, salicylic acid; GA, gibberellic acid.

**Figure 4 ijms-21-02042-f004:**
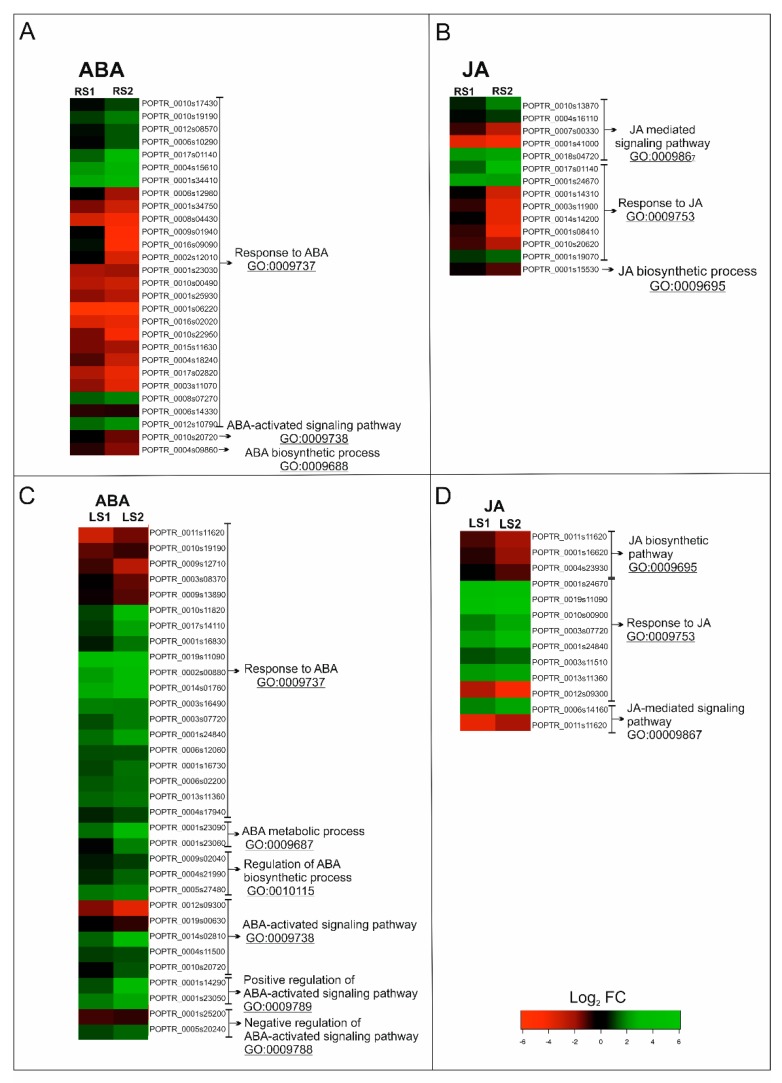
Changes in the expression of phytohormone-related genes in roots (**A**,**B**) and leaves (**C**,**D**) during senescence. (**A**,**C**) Heatmap illustrating the expression profiles of ABA-related genes. (**B**,**D**) Heatmaps illustrating the expression profiles of JA-related genes. Abbreviations: ABA, abscisic acid; JA, jasmonic acid; FC, fold change.

**Figure 5 ijms-21-02042-f005:**
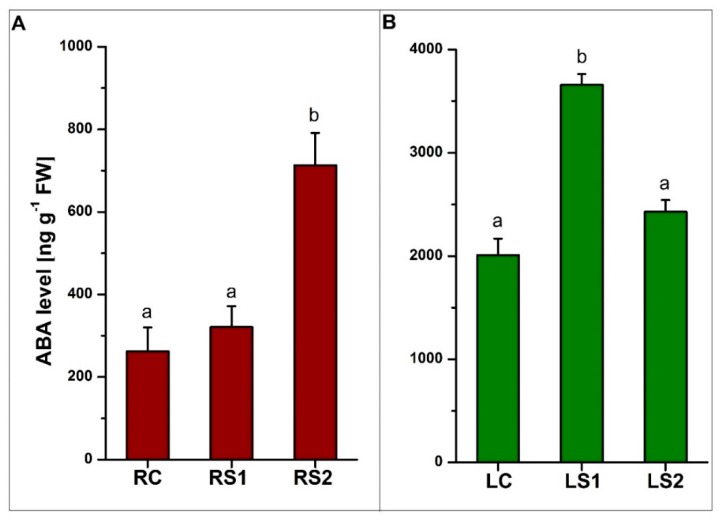
Quantitative analysis of abscisic acid (ABA) levels in fine absorptive roots (**A**) and leaves (**B**) during senescence. Means designated by different letters indicate statistically significant differences according to ANOVA and Tukey’s post hoc test (*p* < 0.05). Values represent the mean ± SE (standard error).

**Figure 6 ijms-21-02042-f006:**
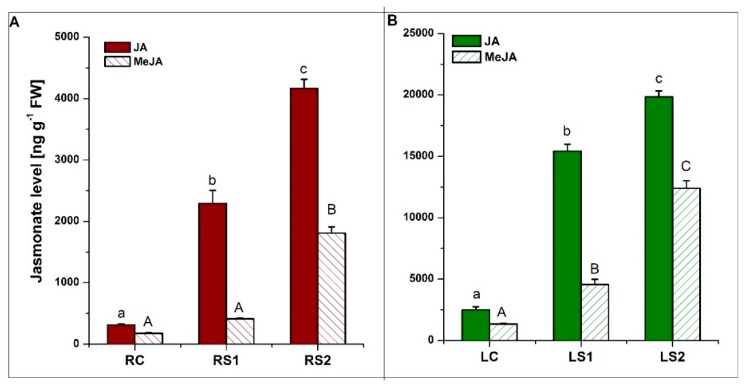
Quantitative analysis of jasmonic acid (JA) and methyl jasmonate (MeJA) levels in fine absorptive roots (**A**) and leaves (**B**) during senescence. Means designated by different letters indicate statistically significant differences according to ANOVA and Tukey’s post hoc test (*p* < 0.05). Values represent the mean ± SE (standard error).

**Figure 7 ijms-21-02042-f007:**
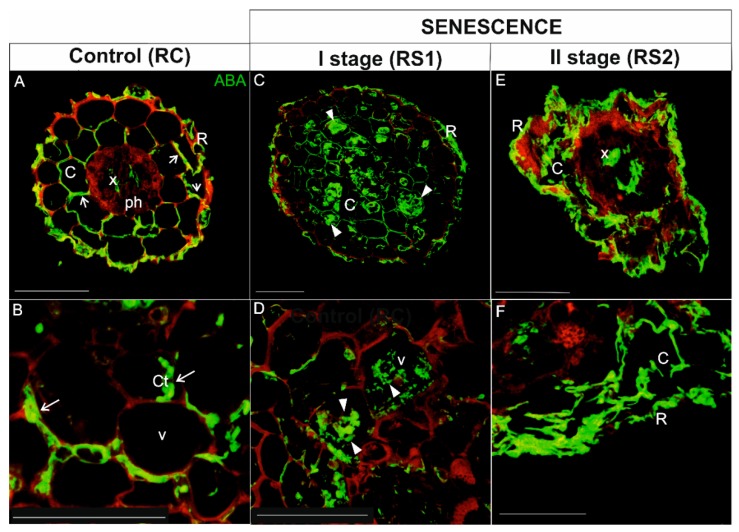
Immunolocalization of ABA (green fluorescence) during senescence process of fine, absorptive roots. (**A**,**B**) Root control (RC); (**C**,**D**) The first stage of root senescence (RS1); (**E**,**F**) The second stage of root senescence (RS2). Autofluorescence (red) of the cell wall was registered in order to visualize the cell/organ shape. Abbreviations: R, rhizodermis; C, parenchyma cortex cells; Ct, cytoplasm; Ph, phloem, X, xylem. Scale bars = 50 µm.

**Figure 8 ijms-21-02042-f008:**
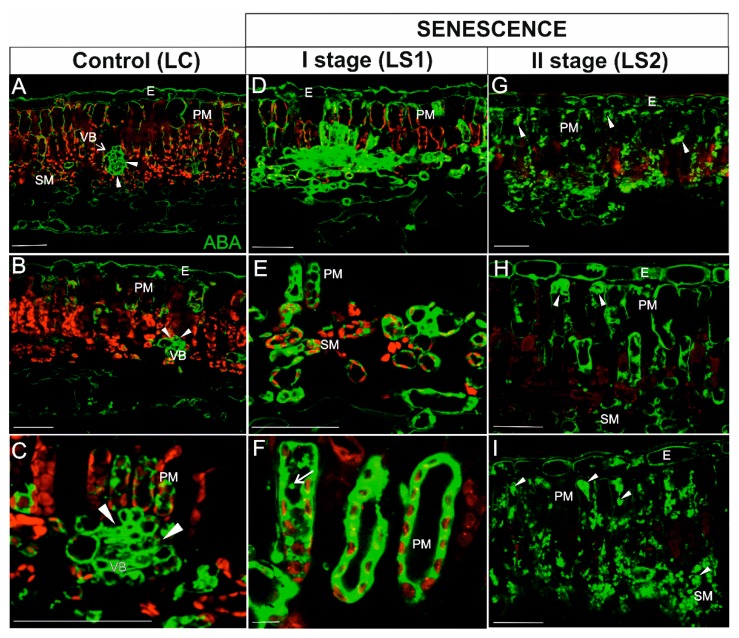
Immunodetection of ABA (green fluorescence) and red autofluorescence of chlorophyll during leaf senescence. (**A**–**C**) Leaves control (LC); (**D**–**F**) The first stage of leaf senescence (LS1); (**G**–**I**), The second stage of leaf senescence (LS2). Abbreviations: VB, vascular bundle; PM, palisade mesophyll; SM, spongy mesophyll; E – epidermis. Scale bars = 50 µm.

**Figure 9 ijms-21-02042-f009:**
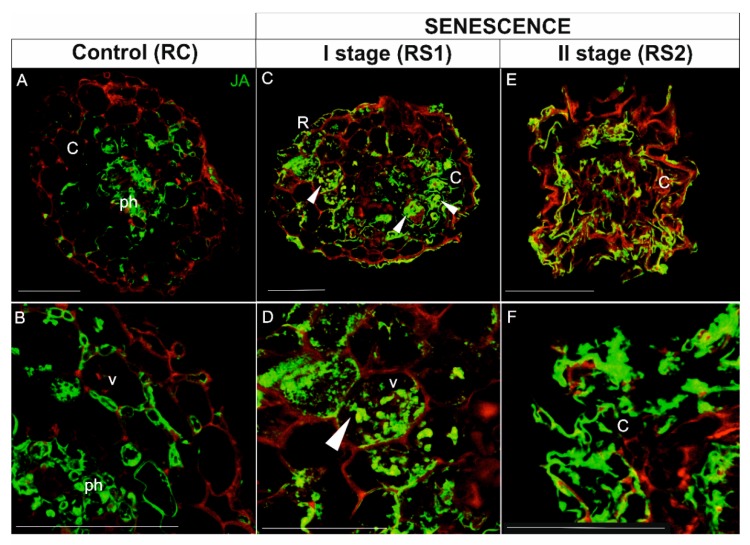
Immunolocalization of JA (green fluorescence) in fine, absorptive roots during senescence. (**A**,**B**), Root control (RC); (**C**,**D**) The first stage of root senescence (RS1); (**E**,**F**) The second stage of root senescence (RS2). Autofluorescence (red) of the cell wall was recorded to visualize the cell and/or organ shape. Abbreviations: R, rhizodermis; C, parenchyma cortex cells; Ph, phloem; X, xylem. Scale bars = 50 µm.

**Figure 10 ijms-21-02042-f010:**
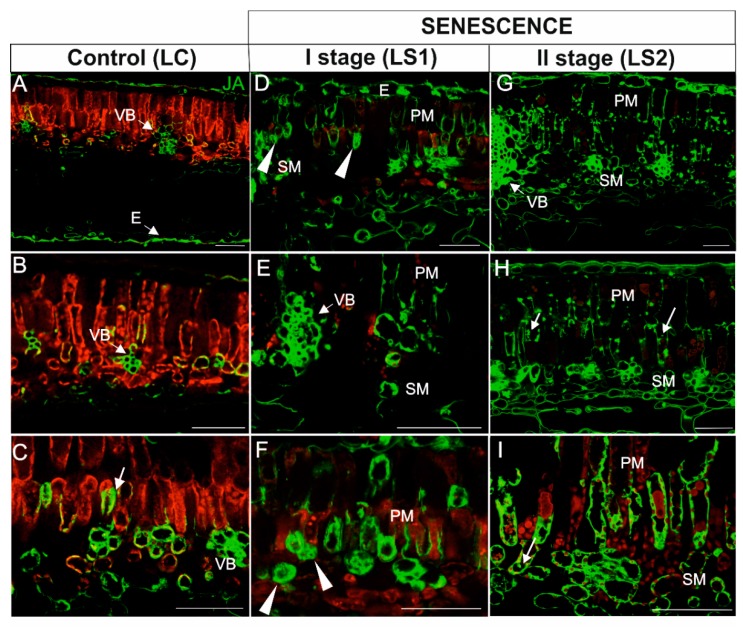
Immunodetection of JA (green fluorescence) and red autofluorescence of chloroplasts in leaves during senescence. (**A**–**C**) Leaves control (LC); (**D**–**F**) The first stage of leaf senescence (LS1); (**G**–**I**), The second stage of leaf senescence (LS2). Abbreviations: VB, vascular bundle; PM, palisade mesophyll; SM, spongy mesophyll; E, epidermis. Scale bars = 50 µm.

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
