# Peer review of "Abscisic Acid and Jasmonate Metabolisms Are Jointly Regulated During Senescence in Roots and Leaves of Populus trichocarpa"

_ijms, 2020, doi:10.3390/ijms21062042_

Round 1
Reviewer 1 Report
The manuscript describes the comparison of leaf and absorptive small root senescence of Populus trichocarpa using two approaches: microarray hybridization of transcripts and the temporal and spatial determination of hormone (ABA, JA) levels. The theme is interesting, and the presented experiments were carefully accomplished. However, there several problems with the presentation and the evaluation of the data. One of the major problems is that the transcriptomic data were not confirmed by other independent methods such as RT-QPCR. It is a must in all transcriptomic experiments. The other main problem is that the expression of known senescence associated genes (SAGs) is not reported in the samples. These could serve as a good endogenous control to show the progression of senescence. It would especially important for the roots, since there is not any evidence that these organs indeed go through senescence and not necrosis. The authors do not give any clues how senescence progresses in the roots, although this is their emphasized aim. Furthermore, the authors themselves discuss that these organs could have been exposed to cold and pathogens beside ageing. If it is really the case, then the comparison of the process in roots an leaves in the presented system is not possible.
Could the COR genes be upregulated due to the passed time and the continuously decreasing temperature between taking the control and RS samples?
There is no web link provided where the crude microarray data are deposited? At least an excell table summarizing all DEGs (with putative functions) should be provided to allow the examination e.g. SAG expression etc. The gene ID-s do not give any clue, and the authors discuss the putative function only few selected DEGs.
There is no direct comparison of leaf and root ABA and JA -related DEGs. Although it is discussed that they hardly overlap, it might be shown e.g. in a Venn diagram or a table.
Other remarks:
The title does not well reflect the content since it rather indicate similarity than dissimilarity of the organs, and do not indicate the transcriptomic approach.
First sentence of Introduction is not correct: senescence is not synchronized but rather a coordinated series of events, moreover senescence, especially organ senescence, is not limited to the end of the plant life cycle.
Line 60: senescence is not a type of PCD. PCD is a cellular process that is often associated with senescence.
Figure 1: wrong sentence in RS2 root image: should be „shrinkage is visible”
Figure 2 part B: wrong labelling as there are two „Upregulated transcripts in RS1”
Figures 7-10. Parts of the figures are labelled with capital letters but in the legend, it is not indicated what the letters mean. It is not described what the pictures int the upper and lower rows show.
Paragraph 265-274 wrongly aligned.
Author Response
We are grateful to the Reviewer whose careful comments have enabled us to make substantial improvements to our manuscript. We feel the paper is been strengthened by the comments and we hope we have met their concerns. We attached the file with the response for all comments. The changes in the main text were labelled in red colour.

Reviewer 2 Report
The paper address an interesting question in the area of senescence: is the senescence of fine roots similarly regulated compared with leaf senescence? The authors used a relevant plant model to address this question, and the research design sounds. However, I have serious problems with the presentation and the description of the results, the reporting of the material and methods section and the overinterpretation of the results in the discussion/conclusion/title of the present manuscript. I think that this work is interesting and could be published by adding some controls and by improving the results and the discussion parts.
Major points:
- The authors have to determine the senescence stage of the studied leaves and roots with quantitative measurements. This control is essential. For example, there is no data about the quantity of chlorophyll in leaves in the figure 1. Considering the leaf heterogeneity in terms of green, how the chlorophyll content has been quantified ? This information is also lacking in the paper Wojciechowska et al., 2018...
- If these samples come from a previously published study, then it must be specified at the beginning of the results paragraph…In this context, a brief summary of these results is expected in order to help/convince the reader about the stage of the samples.
- If these samples are new, then their physiological stage must be deeply described, since the senescence stage is not the same in greenhouse conditions between different experiments. As an example, the authors are arguing that the fine roots of their plants may have experienced cold stress during the senescence process…
The figure 1 must be improved in this way. Please also add a timeline of the season and specify the date of sampling (spring, summer, autumn ?).
- The comparison of gene expression was performed between the three groups “control”, “LS1”, and “LS2” for the leaves and the roots. However, no comparison was done between leaves and roots. Therefore, it is not possible for the reader to determine if some regulations of gene expressions are shared between leaves and roots during senescence. Such information is central for this work and the question addressed by the authors, because this could be the case for ABA and JA related genes. Since the authors briefly mention these results in the discussion (L353-354), a corresponding figure must be included in the present manuscript and must be adequately described in the results section.
- The report on the quantification of phytohormones is lacking several control procedures or which are not clearly detailed in the present study. Does any metabolic standard has been added during the extraction step for the normalization of the results? Does any phytohormone standard has been injected on GS-MS for the absolute quantification of ABA and JA? If the authors used them, this must be mentioned, and if it is not the case for both, then the results are not usable. In addition, the quantity of phytohormones must be reported per gram of dry weight according to the procedure explained in the material and method section. Since a change in the fresh weight/dry weight ratio is expected during senescence, this normalization is crucial to correctly interpret the differences. In fact, what is the fresh weight/dry weight ratio of the samples studied ? again, a physiological description of the samples used in this study is missing (see comment 1).
- There are serious problems with the description of the transcriptomic analysis. The figures related to GO are not shown at all, so please add them. The annotation of the genes presented in the figures 3 and 4 was not adequately reported in these figures and in the material and method section. Please add this information to the respective figures, and explained how the annotation was obtained in the material and method section. I think that you should also perform the annotation by using Arabidopsis genome, since it is a very well-documented model specie. At least, you should do this for the DEG related to ABA and JA metabolism (you can use BLAST or HMMER). For instance, the figures 3 and 4 are very hard to understand/interpret for a scientist specializing in ABA-JA metabolism (“response to ABA” is a very large group that need to be further developed). Which software was used for the heatmap ? because there are serious problems with the log2FC scale bar: in figure 4, the colour seems to be identical between log2FC 2 and 6, or log2FC -6 and -2; in figure 3, the two ends of the scale bar are missing values. To my opinion, the present heatmap prevents the comparison of gene expression, so please change the scale colour (a heatmap() function exists in the free R-studio software, if needed).
- The figures 6 to 10 are very complicated to interpret, perhaps reflecting the absence of a conclusion for this part of the results (see comment 25). I strongly recommend to the authors to only keep the main results in the main text and to move the other parts of experiments that are not essential for the purpose of the study in supplemental figures. The use of bold arrows for example to point out important findings on some pictures (example Fig 9D heterogeneity of JA distribution within the cells) is strongly encouraged.
- The discussion and therefore the conclusion obviously overinterpret the results with respect to the literature. They must be changed to fit the real discoveries of the paper. Please be very careful to avoid scientific shortcuts in the discussion and the conclusion. A correlation between the regulation of gene expressions and the change in ABA and/or JA does not necessarily implies that these genes can effectively regulate ABA and/or JA content at the transcriptional level in Unless, you have already mutants for these genes and you can confirm these conclusions with phenotypic analyses and many experiments? In addition, the fact that ABA and/or JA content increase during senescence in leaves and roots does not necessarily means that ABA and/or JA can promote senescence in populus. Unless you have exogenously applied ABA and/or JA on leaves and roots of populus and confirmed that it can trigger senescence? Please be more consistent and realistic with the discussion of your data.
- The immunolocalization control for roots mentioned in the material and method is missing in the supplementary material 1.
Minor points:
- L37 : Senescence can also occur at the vegetative stage in many crops. Please change the sentence accordingly and add the corresponding references.
- L47-48: The authors do not mention anything about organelles (chloroplasts, mitochondria) during senescence. However, they are central in the remobilization processes during senescence and for the maintenance of a cellular activity during these processes. Since their role in plant senescence had been investigated many times, I strongly recommend adding few sentences with the corresponding references.
- L77-88: This part is clearly lacking for consistency and must be improved in terms of writing style. The authors have to clearly explain what is the interest for comparing leaves and roots senescence. Considering the importance of the chlorophagy in the senescence process, it could be interesting to highlight this difference between leaves and roots. The authors also have to highlight the major conclusions of this work with two brief sentences. Please consider revision for the terms “to report on significant changes that occur in global gene expression”, « In addition to the transcriptomic analysis utilizing a microarray analysis”,” physiological and microscopic analyses 84 were also conducted which provided new information”.
- L91: Please start by introducing your plant material before your microarray analyses. How the plants were grown ? Does the senescence occur at the vegetative or reproductive stage ? You could start with something like “In order to compare changes in gene expression between leaves and roots senescence, plants at the vegetative stage were harvested in XXX …etc”.
- Figure 1: There is no data about the quantity of chlorophyll in leaves. Considering the leaf heterogeneity in terms of green, how the chlorophyll content has been quantified ? This information is also lacking in the paper Wojciechowska et al., 2018… The authors have to prove the senescence stage of the studied leaves and roots with quantitative measurements. If these samples come from a previously published study, then it must be specified at the beginning of the results paragraph…
- L99: Please also mention the correction used for the ANOVA (even if it is already mentioned in the material and method section). Considering the comparison performed
- Figure 2: There is an error in the figure 2B (Up-regulated transcript in RS1 is mentioned two times. The legend for the meaning of the colours is also missing for the entire figure. In addition, a transcript cannot be “up-regulated” whereas a gene expression can be. Please change the terms “up or down-regulated transcripts” with something more accurate (“over or underexpressed genes” perhaps). In order to facilitate the interpretation of the results: please add the items “RS1/Control” and “RS2/control” to the graphs 2A and 2C, add a main title for the “leaf part” and the “roots part” in the figure, change the colours of the Venn diagram to make it more visually interpretable (there is a free R-package called “VennDiagram” if needed).
- L114-120: The description of these results does not refer to any figure. Can you add a figure with the corresponding results please ? (main text or supplemental data).
- L129-137: Again, the GO data are missing in the figure 4A. Please add them
- Figure 3 and 4: there is a big problem between the text and these figures. The figure 4A is mentioned before the figure 3B and 3C which is very confusing for the readers…I suggest to merge Figure 3A and 4A into one figure and to merge the figure 3B, 3C, 4B and 4C into another figure that will be only focused on ABA and JA related genes. In addition, since green and red colours are often referring to up or down-regulation of gene expression in the paper, please change the colours for the pie charts in figure 3A and 4A. The figure 3B, 3C, 4B and 4C should mention the nearest homolog to each populus gene, by using Arabidopsis genome as a reference, in order to help the reader to understand the importance of the DEG reported. Please add the meaning of “FC” in the corresponding legends.
- L153-200: The authors must explain the role of some ABA or JA-related genes that were identified to be differentially regulated. For example, some ABA-related genes encoding aquaporins-related proteins were down-regulated in roots, but we do not know the importance of those genes in the ABA signalisation….I do not expect a discussion of the results but at least the authors should mention the importance of some results observed in the transcriptomic analysis, considering the ABA and JA signalling pathway.
- L204-205 : A brief summary of the findings from the transcriptomic analysis is expected, as well as a transition sentence to explain the interest of quantifying ABA, JA and meJA hormones.
- Figure 5 and 6: The legend mention a post-hoc tukey test but not the first test that has been performed (ANOVA maybe ?). Please add the name of the first test also. The authors must briefly explain the system used for the quantification of the phytohormones and must add the number of biological replicates in the legend of these figures. For the figure 5B, from the standard error presented on the bar graph, I seriously doubt about the statistical difference between LC and LS2 by using an ANOVA-Tukey HSD (p<0.05). As expected for metabolomics data, please provide the full dataset for hormone quantification as supplemental data. The ratio between JA and meJA is not described/discuss at all in the result section. Please add a corresponding bar graph and few sentences.
- L292-295:This short summary reflect the results part: the authors do not clearly conclude about the major findings of this section. They must summarize briefly their results and their correlations (transcripts levels-metabolites-cellular localization)
- L304-305: “Leaves as a nutrient generating organ” does not make sense. DNA, lipids, organic and amino acids can be synthesized in roots also… I think that leaves were studied because it is easy to monitore the onset of senescence with visual inspection (yellowing), whereas it is not the case for roots.
- L318-319: See comment 14-15
- L331: Impossible to suggest that with the experiments presented. To conclude this, you need at least to prove that exogenously applied ABA on fine roots can trigger their senescence. Here, with your results, we can only suggest that “ABA might also contribute to senescence processes”. But its role senescence-promotive role may not be exactly similar between leaves and roots
- L346-347: In my opinion, ABA colocalizes with senescence associated vesicles. You should discuss this results with the relevant literature.
- L349-350: Please add additional references for the role of ABA in chlorophagy.
- L360: I am sure there are plenty of transcriptomic analysis that have been performed on many plants during leaf senescence since 2006…Can you please add recent references?
- L367: Again, you cannot conclude this from your data. The mechanism in Arabidopsis is not necessarily the same in populus, only because you found similar regulation of some genes sharing a moderate protein sequence homology. To prove that, you should analyse the phenotype of the mutants for your genes of interest in populus. Here, you can say “Collectively, the data suggested that the PP2C identified in our transcriptome analysis could be involved in the ABA-dependent regulation of leaf senescence”.
- L387-388: You cannot prove the importance of the role of ABA in senescence with your experiments…You can only say that “ABA content is tightly regulated during senescence in leaves and roots, possibly at the transcriptional level, and its accumulation may contribute to senescence processes in both organs”.
- L389-392: Have you proved that ABA increase cold tolerance in fine roots of populus with a transcriptional regulation, in your experiments ? I don’t think so. Please stop to overinterpret the results…
- L402-408: Here, it is exactly the opposite, as you underinterpret your results…You should say that “variations of JA content in roots of populus is a metabolic signature of senescence” and explain why it is new (no one works with roots) and how it is different or not from previous analysis with different organs ! Of course, you cannot conclude that JA promote or not senescence in roots of populus.
- L414-418: Have you identified or quantified the presence of microorganisms in your RS2 samples from the present study? Therefore, you should remove this paragraph as it cannot be relied to your experiments. Perhaps, just say that “accumulation of JA in roots may be related to multiple factors independent of senescence such as : plant immunity, transport processes..etc”
- L439-444: How can you conclude that JA could be involved in the response of biotic stress in senescing roots with your experiments ? Overinterpretation !
- Supplementary material 2 and 3 are not cited in the text. Please add them. Add the meaning of “FC” and the test used for statistics on DEGs in the legend. Fig supp2 “ethylene”
- Please change the title. It must fit the major conclusions of the paper. For instance, your paper does not prove that ABA and JA are involved in the regulation of senescence in roots and leaves. However, your paper prove that “ABA and JA metabolisms are jointly regulated during senescence of roots and leaves of populus trichocarpa”.
List of typos I have spotted:
L56-57: The sentence is not written correctly in English. Please consider revision and replace “;” by “,”.
L169 : « Among the DEGs with increased expression » -> Among the genes up-regulated by senescence ?
L179 : « encoding an (POPTR_0009s13890) and » - not clear
L209: Please rephrase this sentence…
L310: mechanisms THAT have
L328: “notably” or “however” but not both of them to start a sentence.
Author Response

(The authors gave the same response as above.)

Round 2
Reviewer 1 Report
My questions were answered and the required corrections were made.
Author Response
We are grateful to the reviewer for all previous comments that allowed us to significantly improve our manuscript. We are glad that the Reviewer has accepted all changes and responses to comments.
Reviewer 2 Report
I thank the authors for the modifications that helped to improve the reading of the paper. However, I have still few points to raise:
- Nothing has been added at the beginning of the result part to introduce the biological sample ! For the second time, please add few sentences to explain to the reader that the samples come from another publication, in which the senescence stage was deeply characterized with chlorophyll measurements, morphological symptoms etc… with a reference for the material and method section. The timeline added in the figure 1 is not a true timeline, as there are no date… Add the date of sampling please. We need to know the true time interval between each sampling. Concerning Wojciechowska et al., 2018, there are no standard deviation for the chlorophyll measurements. Considering the leaf and plant heterogeneity, there must be a standard deviation of around 5% at least… Do the authors can provide us an explanation for this ethically critical problem?
- Upon second reading of the figure 2, I don’t understand what is the interest of the figure A and C, since the same information is presented in the figure B and D. I would suggest to remove the figure 2 A and C and to add Venn diagram that compare leaves and roots at the three sampling date (which has not been performed in the response of the authors...). But the authors can continue to argue that there are a lot of charts at work, even if it is obvious that the complete analysis of the dataset has not been performed, or is not shown…
- Considering the calculation of ABA on a dry weight basis: I don’t know what has been calculated by the authors but it is definitely wrong….with a FW/DW ratio of 4, the ABA content per FW cannot be multiplied by a factor 33…The recalculations are very doubtful…Nevertheless, the fact that the FW/DW ratio is unchanged during senescence is very interesting considering the literature. It should be included in supplemental data, and cited when the authors describe their biological samples in the beginning of the result part.
- I have a problem with the overinterpretation in the abstract. L30-33: your results have not proved that !!! You should be more consistent with your results and only “suggest” conclusions, when they cannot be fully drawn based on your results. Here is a possible example: “ We have shown that the regulation of ABA and JA metabolism is tightly regulated during senescence processes in both leaves and roots. The results were discussed with respect to the role of ABA in cold tolerance and the role of JA in resistance to pathogens”.
Author Response
We are grateful to the Reviewer who careful comments have enabled us to make substantial improvements to our manuscript. We feel the paper is been strengthened by the comments and we hope we have met their concerns. Detailed indications which parts were modified along with the response for Reviewer comments are attached.
